# Wearable Systems for Unveiling Collective Intelligence in Clinical Settings

**DOI:** 10.3390/s23249777

**Published:** 2023-12-12

**Authors:** Martina Pulcinelli, Mariangela Pinnelli, Carlo Massaroni, Daniela Lo Presti, Giancarlo Fortino, Emiliano Schena

**Affiliations:** 1Research Unit of Measurements and Biomedical Instrumentation, Department of Engineering, Università Campus Bio-Medico di Roma, Via Alvaro del Portillo 21, 00128 Roma, Italy; martina.pulcinelli@unicampus.it (M.P.); m.pinnelli@unicampus.it (M.P.); c.massaroni@unicampus.it (C.M.); e.schena@unicampus.it (E.S.); 2Fondazione Policlinico Universitario Campus Bio-Medico, Via Alvaro del Portillo, 200, 00128 Roma, Italy; 3DIMES, University of Calabria, Via P. Bucci 41C, 87036 Rende, Italy; giancarlo.fortino@unical.it

**Keywords:** wearable systems, assessment of collective intelligence, physiological parameters estimation, assessing group behaviors

## Abstract

Nowadays, there is an ever-growing interest in assessing the collective intelligence (CI) of a team in a wide range of scenarios, thanks to its potential in enhancing teamwork and group performance. Recently, special attention has been devoted on the clinical setting, where breakdowns in teamwork, leadership, and communication can lead to adverse events, compromising patient safety. So far, researchers have mostly relied on surveys to study human behavior and group dynamics; however, this method is ineffective. In contrast, a promising solution to monitor behavioral and individual features that are reflective of CI is represented by wearable technologies. To date, the field of CI assessment still appears unstructured; therefore, the aim of this narrative review is to provide a detailed overview of the main group and individual parameters that can be monitored to evaluate CI in clinical settings, together with the wearables either already used to assess them or that have the potential to be applied in this scenario. The working principles, advantages, and disadvantages of each device are introduced in order to try to bring order in this field and provide a guide for future CI investigations in medical contexts.

## 1. Introduction

‘Collective Intelligence’ (CI) is a typology of intelligence that emerges from the collaborative endeavors of numerous individuals who are capable of engaging in intellectual cooperation aimed at creating, innovating, and inventing. CI is known as the general capacity of a group to effectively undertake a diverse range of tasks [1]. CI is a property of the group itself and has been found to possess predictive strength that surpasses that which can be revealed by only considering the individual capabilities of its members [2].

The concept of measurable human intelligence emerged for the first time in 1904 with the studies of Spearman, who defined general intelligence, g [3], which arose from the observation that those who excel in one particular task also exhibit high performance in various other tasks. Statistical methods were employed for measuring g. Nevertheless, at the beginning, nobody had methodically investigated if a comparable form of intelligence exists for groups of individuals. Only recently has the idea that intelligence is not merely something that occurs inside of individual brains started to gain traction.

According to [1], Woolley and colleagues exploited the same statistical techniques used in the individual intelligence tests to create group intelligence tests. Participants had different kinds of tasks to carry out as a team and the results show that, even for a group, there is a single statistical factor that demonstrates the capacity of the group to succeed in different types of tasks. What emerges from this study is that the members’ skills, the averaged individual intelligence, and the maximum individual intelligence of people in the group influence the CI, but only moderately [2]. CI is not strongly correlated with these factors; in fact, having a group of smart people is not enough, alone, to make a smart group. On the contrary, CI was shown to be significantly correlated with the mean social sensitivity of participants, the level of diversity in the team, and the equitable distribution of speaking turns [2]. It was discovered that, if one person dominates the conversation, the group results in being, on average, less intelligent than a group in which people take more turns talking. These preliminary studies paved the way for further research in this field and the number of CI investigations is experiencing a substantial growth. This is because, if maximized, CI allows people to solve issues and carry out tasks by working in groups and getting beyond individual limitations [2,4]. 

In particular, CI is becoming increasingly prominent within the healthcare sector thanks to its capacity to enhance cooperation, teamwork, and patient care through more effective medical treatments [5]. There is a strong belief that teamwork is a crucial element in healthcare delivery [6]. Various studies have highlighted its positive impact on performance outcomes in several scenarios such as operating rooms (ORs), intensive care units, and nursing homes [7,8,9,10,11]. A deficiency in teamwork is frequently pinpointed as a vulnerability that can compromise the quality and safety of healthcare services [6]. Research conducted within ORs has uncovered that inadequacies in teamwork, leadership, communication, and suboptimal decision-making are not uncommon and can result in adverse consequences [12].

At the moment, there is evidence supporting the existence of CI, but there are not yet theories capable of explaining how to measure CI. What is clear is that at the basis of CI study lies the assessment of team dynamics and social interactions, taking into account both individual and group aspects that, together, contribute to influence team outcomes. Nowadays, researchers still often rely on surveys and human observers for studying group dynamics. These methods allow for investigating people’s attitudes for leadership, communication, teamwork, stress, and fatigue, as well as the organizational capacity of each member. For instance, in some studies [12,13], the Operating Room Management Attitudes Questionnaire (ORMAQ) has been introduced to offer valuable diagnostic insights concerning behavioral patterns and safety aspects within surgical teams. It has been used to find out areas where clinical staff have found greater cognitive overload in performing technical activities. In this scenario, the Observational Teamwork Assessment for Surgery (OTAS) seeks to be an all-embracing analysis of teamwork in OR. The OTAS assesses the teamwork behaviors of all the OR team members simultaneously, unlike other existing approaches that consider only one medical profession [14,15].

Though useful, an approach based on surveys and self-reports has unluckily resulted in being inaccurate, time-consuming, and non-exhaustive [16]. In fact, direct human-to-human observation is costly, does not allow data collection from many subjects, and there may be inconsistencies among observers’ assessments. In contrast, the automated capture of social interactions, by measuring both individual and group parameters, can represent a valuable alternative.

In particular, it can facilitate data gathering from sizable populations and it has the potential to mitigate some current limitations, such as constraints on the scale and frequency of surveys [17], without disturbing or slowing down the work of medical teams.

In the literature, the group parameters monitored for studying the social and organizational dynamics include the number of face-to-face interactions (F2Fs), proximity time, and speaking time, while the individual ones are the heart rate (HR), heart rate variability (HRV), respiratory rate (RR), galvanic skin response (GSR), and physical activity level [17,18,19,20].

Recent technological advancements in smart sensing have opened new possibilities for implementing innovative systems capable of monitoring group and individual metrics. Wearable systems are emerging as innovative technological solutions suitable for analyzing individual traits and human health in various fields ranging from sport to clinical settings without impairing the user activity. The need for automated measurements to unveil CI in medical settings can benefit from the use of wearables. However, only a few studies focused on the use of wearable devices (e.g., smart computing devices, smart clothing, GPS tracking, videos, Google Glasses, RFID, etc. [5,17]) in this scenario [5], due to the criticality of the application context; no one has yet employed different wearable technologies to simultaneously monitor the complete set of mentioned parameters.

For instance, in [19], all the abovementioned group parameters together with the physical activity level were monitored by wearable electronic sensors (i.e., the Sociometric Badges) to evaluate the performance of a post-anaesthesia care unit by examining all nurses working together in the unit [18]. Although the Sociometric Badges represent promising technologies for detecting characteristic parameters to unveil CI, they do not allow an analysis of all the metrics of interest and the physiological signals remain excluded. The latter can be useful to evaluate the individual stress and cognitive workload of medical team members and, consequently, to establish an effective teamwork model, since it is known that they can be used to prevent stress-related diseases [18,20,21,22,23,24]. 

Many studies have proposed smart clothing to detect stress and cognitive workload from HR, HRV, RR, and GSR, with the aim of assessing how each member of the team reacts to unforeseen events [25,26,27]. 

There is a need of a comprehensive review of all the features that can be studied and monitored for evaluating social interactions to unveil CI. This narrative review focuses on the main individual and group parameters that are reflective of CI, emphasizing the potential of wearables used for assessing them. The working principles, advantages, and disadvantages of each system will be outlined with the intent of offering a guide for future CI investigations. 

In more detail, this study intends to converge these assessments in a clinical scenario, where an accurate analysis of the behavioral dynamics and the psychophysical state of surgeons, doctors, and nurses is crucial [28].

## 2. Wearables for Monitoring Group Behaviors

Over the years, researchers have endeavored to delineate and analyze personality traits and individual predispositions to respond emotionally or behaviorally in distinct manners. Groups are seen as collections of individuals, and group behavior is considered as an outcome of the scaling up of individual behaviors to the group level [29].

In terms of CI, assessing team members’ behavioral parameters appears necessary for evaluating the group efficiency. Currently, studies have focused on the number of F2Fs, the speech energy, the percentage of speaking time, and the proximity time [18].

Below is an analysis of the main group metrics that are considered relevant for characterizing CI. The importance of each parameter will be highlighted, along with the wearable technologies used to evaluate each one, with a focus on the clinical setting. In particular, the sensors embedded into wearables and their body locations are shown in Figure 1. 

A schematic overview of the primary studies focusing on the use of wearables for monitoring group parameters is reported in Table 1.

### 2.1. Face-to-Face Interactions

A significant number of nonverbal cues is conveyed via F2Fs: face expressions, voice features like tone and volume, glances, movement mirroring, level of engagement, dominance of the conversation, and so on [20].

F2F is a rich medium because it offers a variety of social indicators through body language and natural language, which considerably reduces conversational ambiguity. According to this concept, F2Fs are most successful for explaining confusing events and creating a common understanding when the information to be communicated is ambiguous or uncertain [30]; F2F is often the most expensive (in terms of time, effort, and energy), yet it is favored for imparting complex knowledge since it can help clear up misunderstandings.

For this reason, the analysis of this parameter is highly useful in the clinical scenario; F2Fs between the team members are involved in initial team briefings but are especially relevant during the operational phase [28]. The existing literature strongly suggests the need for F2Fs in clinical contexts, including critical settings. Nevertheless, the absence of objective techniques and inadequate datasets poses significant constraints [20]. Effective cooperation and collaboration are valued generally, but there is little specific research on F2Fs. In the healthcare industry, cooperation and communication are linked to staff happiness, medical costs, and patient care quality [31]. Moreover, empiric studies show that F2F is correlated with effective teamwork and decision making, particularly in high-pressure scenarios [20].

From the literature, it has emerged that the wearables devices for the detection of F2Fs are mainly based on infrared (IR) sensors and Radio Frequency Identification (RFID) technology [17,18,20,22,32,33]. 

In general, the following features can be computed: the total F2F time per person, the number of F2Fs, and the amount of different people with F2Fs [17,29,34]. In some cases, the starting time and durations of each F2F are also registered [22]. Moreover, the identification of the faced partner may occur by leveraging the different IDs detected [22]. 

Considering the IR transmissions, a F2F occurs whenever a receiving sensor detects an IR signal. Based on this technology, the features that can be obtained to characterize F2Fs are defined as follows. In [29], the number of F2Fs is evaluated by dividing the IR detections per minute by the IR transmission rate. According to [17], the total F2F time per person can be determined by multiplying the amount of consecutive IR detections by the IR transmission rate.

In [22], a wearable system with IR communication to measure the starting time and duration of F2Fs with each partner is introduced. To assess the functionality and potential of the developed system, they carried out tests in a school for children with intellectual disabilities and/or autism spectrum disorder (ASD). It is represented by a smart device worn on the forehead like a headband. The IR module consists of an IR receiver and an IR LED. One is responsible for modulating data, including its own ID, and sending them via an IR LED to another system that detects the IR light, demodulates the signal, and subsequently transmits it to the microcontroller. In this way, this technology also enables the identification of the faced partner. The communication between transmitting and receiving devices can be established only when both are positioned on the user’s forehead with the light direction precisely aligned with the user’s face. The wearable works with different states: “standby”, when it remains in readiness, awaiting signals from other systems, “being faced”, when only one of the two is gazing at the other, and finally “face-to-face” if they both face each other. The introduced device results in being easy to wear and lightweight, so that it avoids limiting the user’s movements. It allows for real-time measurements and provides visual feedback. Despite the promising features, the main drawbacks are that it cannot precisely detect F2F when the device is not aligned with the user’s facial direction and the proper detection depends on the distance between users. 

Moreover, other typologies of wearables based on an IR sensor for detecting F2Fs are the sociometric wearable devices (SWDs) [20]. In particular, they are typically worn around the neck and designed to automatically record some individual and collective parameters (e.g., F2Fs, speaking time, physical activity level, and proximity time), by using different sensors embedded in the same device. These systems have also been applied in the clinical field, more precisely in post-anaesthesia care units (PACUs), surgical units, and intensive care units (ICUs) [16,18,35,36,37,38,39,40,41,42]. The Business Microscope [32,33] is a type of SWD that includes a three-axis accelerometer and IR sensors. In particular, the badge is equipped with six IR transceivers positioned on its front side, each oriented at different angles to detect F2Fs.

Another commonly used SWD is the Sociometric Badge [17,18,19], which was proposed by MIT researchers. It comprises four main elements: an IR sensor, a three-axis accelerometer, a microphone, and a Bluetooth module.

Both the introduced SWDs result in being easy to wear and lightweight, avoiding limitations in user’s movements and allowing the identification of the faced partner, which are important features for employing them in clinic settings. However, they have some drawbacks: they present an obstruction issue, they can only detect the F2F of individuals wearing the devices, and they do not return visual feedback. Moreover, no F2F results if the involved subjects are outside the range of IR communication. Particularly for the Sociometric Badge, it is essential that the two IR transceivers have an unobstructed line of sight, and the receiving sensor is placed inside the transmitter’s IR signal cone. This cone must adhere to specific constraints for both height (h) and radius (r), which are as follows: h ≤ 1 m and r ≤ h × tan θ, with θ ≤ ± 15° [36]. As for the Business Microscope, F2F is detected only if the badges are within a horizontal angle of 120°, a vertical angle of 60°, and a radius of 2 m [32].

Other systems used to detect the considered parameter are represented by SWDs, which rely on Radio Frequency Identification (RFID) technology [23,43]. They have been applied in clinic scenarios (e.g., in a general pediatrics hospital ward [24]), but also in many others, such as conferences [23] and schools [44]. They are RFID badges, worn around the participant’s neck or on the chest, that are able to detect the F2F interactions and spatial proximity of individuals through the exchange of low-power radio packets between tags. 

As well as the abovementioned SWDs, this typology of devices also does not return visual feedback and can only detect the F2F of individuals wearing the devices. No F2F is detected if the involved participants are outside the communication range between the devices, which approximately ranges from 1 m to 1.5 m [24].

The measurement quality can be affected by the communication distance, body angles, movements, number of participants, hand gestures, and the presence of other objects [43]. Although the measurement’s sensitivity and specificity using RFID technology are not particularly high, they might rise to a reasonable level if there is at least 75 s between two consecutive F2F interactions [43].

### 2.2. Proximity 

A group members’ physical, social, and cognitive distances from one another are referred to as proximity. Team member closeness relates to features of the teamwork quality construct, such as mutual support, high-effort work norms, and cohesiveness [45]. Researchers can learn more about how people interact and how their interactions effect group results by assessing proximity. Social proximity, such as shared identity or similar interests, can enhance communication and collaboration, leading to stronger group cohesion and higher levels of trust, whereas physical closeness can boost the possibility of communication and collaboration among group members, which can improve their ability to solve problems collectively, according to research. Therefore, on the basis that a smaller physical distance enhances the likelihood of interaction and information sharing, several studies have anticipated that close proximity between team members will boost team collaboration [20,45,46,47,48].

Recently, a slew of proximity-sensing technologies, such as IR sensors, wearable radio frequency (RF) devices, Bluetooth-enabled smartphones, radio beacons, and others, have facilitated the automatic gathering of proximity data. 

One innovative method of proximity sensing uses an oscillating magnetic field [49,50]. Due to the special characteristics of the magnetic field, this technology is dependable and excels at differentiating between distances greater than and less than 2 m.

Proximity-sensing wearable devices (henceforth called “tags”) in particular represent cost-effective solutions, possess minimal operational limits, and can be tailored to assess proximity data every few seconds [23,47,51,52,53]. They gather information on temporal dynamic interaction patterns when people wear them in real-world environments, including classrooms, hospitals, and conferences. As a matter of fact, wireless sensor network technology permits the reconstruction of the social network relevant for infectious disease transmission [54] by assessing F2F proximity with high levels of spatial and temporal resolution [24,55].

As previously noted, in the measurement of F2Fs, recent research seeks to establish a positive correlation between SWD data and medical outcomes. SWDs are specifically engineered to record and collect data on human interactions, and they possess the capability of automatically capturing close proximity by measuring who interacts with whom, for how long, and at what location [20,55,56].

Embedded IR sensors detect the proximity of other people who are wearing sociometric badges by analyzing the strength of the reflected IR signal within a certain range. The data from the IR sensors can then be combined with other data from the badge, such as movement and location data, to provide insights into social interactions and patterns of behavior [33].

Meanwhile, RFID technologies have raised significant expectations for their applications in clinical settings. They stand out as the prevalent commercial approach for indoor tracking purposes [55,56,57,58,59]. In [55], the proximity information is communicated to radio receivers placed throughout the hospital ward (bedrooms, offices, and the hall). 

In most cases, they detect F2F close proximity, since the ultra-low-power radiofrequency used cannot be transmitted through human bodies [53]. These devices are capable of assessing proximity only among individuals who wear the systems, and no detection occurs if the involved participants are outside the communication range between the badges (i.e., from approximately 1 m to 1.5 m [24]).

Proximity interaction can be also quantified using Bluetooth technology [56,60]; Bluetooth Low Energy (BLE) technology serves as another significant technique for indoor tracking [61,62], especially in confined spaces [63]. This is especially valuable for monitoring clinical workflow, where the need for a reliable real-time location system (RTLS) to monitor patients and clinical staff is pivotal. Because BLE tags and detectors are widely available, this method is adaptable and can be employed in conjunction with a variety of hardware so as to regulate their biases [64]; location data may be gathered in a variety of different places. In most applications, the Sociometric badges are used, which contain an embedded Bluetooth module [35,39,40,56]. They result in being comfortable, easy to wear, and unobtrusive, allowing for real-time measurement. However, the reliability of the proximity detection depends on the distance between devices, especially as it decreases with increasing distance.

As previously mentioned, SWDs hold promise as valuable devices for detecting potential routes of infectious disease transmission in clinical scenarios, which will aid in the creation of infection control plans [54,65]. Several studies have revealed that these kind of technologies can be helpful not only for contact tracing, but also for enhancing users’ situational awareness, for example via alerts and quick inputs regarding their physical distance from another person [66].

### 2.3. Speaking Time

Effective communication is an essential aspect of any successful team or organization, and verbal communication plays a crucial role in facilitating CI. Verbal communication enables team members to exchange ideas, clarify concepts, and build shared meaning. Through dialogue and discussion, team members can identify assumptions, expose blind spots, and challenge each other’s thinking. Moreover, verbal communication enables individuals to convey emotions, build rapport, and establish trust, which are critical for effective collaboration [6,17,18,29].

Researchers state that team efficiency is significantly influenced by oral communication and the amount of time spent in fostering social connections, which cultivates reliance and bonding [67,68,69]. The timing, volume, and changes in speech can be measured [70], resulting in some of the predictive features of group performance. 

MEMS microphones that can be embedded into wearable devices are widely spread, as in Sociometric Badges, which play an important role in enabling the measurement of verbal communication between team members [37]. To protect anonymity, the Sociometric badges can extract speech characteristics without recording the substance of the conversation, and they can wirelessly upload data to a central server. The Badges incorporate a microphone to record vocal signals with a specific sampling frequency. It is important to note that the content of the conversation is not analyzed. In [29], it is established that the sampling frequency (fs) should be set at a minimum of 8000 Hz. This choice is informed by the fact that the frequency band of vocal signals typically spans from approximately 300 to 3400 Hz. For adult males, the fundamental frequency typically falls within the range of 85 to 155 Hz, whereas for adult females it generally varies from 165 to 255 Hz. It has been demonstrated that a number of band-pass filter and speech recognition front-end systems are efficient in detecting speech. Additionally, the notion of determining the speech volume modulation from the filter’s output is related to the fact that here is where much of the speaking energy is located. 

One notable development is the integration of Speech-to-Text Technology, which enables the conversion of spoken words into text, is that this technology, when incorporated into wearables, offers the ability to automatically transcribe conversations. It not only quantifies speaking time but also delves into the content of discussions, providing valuable insights into the nature of interactions within the group [71].

Most of the features that are extracted from the speech activity relate to the speaking time, the overlapping speech time, the total speaking time, which is obtained by considering both the speaking time and the speech overlap time for each participant, and, finally, the dominant speaker (i.e., the speaker who talks for the longest amount of time) [35].

Members of efficient groups tend to engage in more communication, while also maintaining their physical activity levels with minimal variation. They exhibit lower speech energy, interacting closely with their teammates [18].

In a clinical setting, where accurate diagnosis and timely treatment are critical, effective communication and collaboration among healthcare providers are essential. Research has shown that teams that communicate more frequently and effectively are more likely to achieve better patient outcomes, and that speaking time is a reliable indicator of the quality of communication and the degree of cooperation within the team [5,72,73,74,75]. 

Moreover, speaking time can also be used to identify potential communication barriers or biases that may hinder effective collaboration. For example, if certain team members consistently dominate the conversation while others remain silent, it may indicate a power imbalance or lack of psychological safety within the team; in [35], by means of Sociometers, participants were given instructions to engage in a conversation with varying levels of interruption for identifying disparities in vocal overlap. The device proved to be effective in distinguishing the dominant speaker.

In actual clinical environments, residents frequently find themselves having to blend their technical and communicative abilities, while performing tasks like catheter insertion, wound suturing, or invasive examinations. However, completing procedures by developing an empathic relationship with disoriented or upset patients proves to be quite a challenging undertaking [76].

Therefore, measuring and analyzing speaking time can be a useful tool for clinical teams to assess their communication and collaboration and identify areas for improvement. By ensuring that all group participants have an equal chance of speaking and actively participating in the discussion, teams can enhance their CI and ultimately improve patient outcomes.

**Table 1 sensors-23-09777-t001:** A schematic overview of the primary studies focusing on the use of wearables for monitoring group behaviors.

Paper	Parameter	Working Principle	Wearable	Scenario	Pros	Cons
Hachisu T. et al., 2018 [22]	F2F: starting time and duration of each F2F	IR sensor	FaceLooks: headband-type wearable device ^1^	Children with intellectual disabilities and/or ASD	Easy-to-wear, lightweight, real-time measurement, visual feedback, identification of faced partner	Obstructions F2F is detected only if the light direction is aligned with the user’s face
Olguin D. et al., 2009 [36]	F2F: T-F2F ^2^ per person and NP-F2F ^3^	IR sensor	Sociometric Badge, worn around the neck ^1^	Nurses of a post-anaesthesia care unit (PACU)	Easy-to-wear, lightweight, real-time measurement, identification of faced partner	No visual feedback Obstructions F2F is detected only if the receiving sensor is placed inside the transmitter’s IR signal cone (h ≤ 1 m and r ≤ h × tan θ, with θ ≤ ±15°)
Kawamoto E. et al., 2020 [41]	F2F: T-F2F ^2^ per person	IR sensor	The Business Microscope: wearable badge, attached to the participants’ front pockets ^1^	Staff members of an intensive care unit (ICU)	Easy-to-wear, lightweight, real-time measurement, identification of faced partner	No visual feedback Obstructions
Yu D. et al., 2016 [35]	F2F: T-F2F ^2^ for each actor pair	IR sensor	Sociometric Badge, worn around the neck ^1^	Simulated team communication and patient care scenarios at an emergency department’s pediatric ward	Easy-to-wear, lightweight, real-time measurement, identification of faced partner	No visual feedback Obstructions F2F is detected only if the badges are within 2 m at an angle of 30° No F2F is detected during sit/stand combination
Yu et al., 2015 [56]	F2F: T-F2F ^2^ for each actor pair	IR sensor	Sociometer Badge, worn around the neck ^1^	Simulated hand-off scenarios at an emergency care environment	Easy-to-wear, lightweight, real-time measurement, identification of faced partner	No visual feedback Obstructions F2F is detected only if the badges are within 2 m at an angle of 30°
Isella et al., 2011 [24]	F2F: N-F2F ^4^	RFID technology: exchanging of low-power radio packets	Active RFID badge ^1^	Health care personnel, patients, and their caregivers at the pediatric ward of a hospital to study infectious disease transmissions	Comfortable, unobtrusive High-resolution data recordings	No visual feedback F2F is detected only if the badges are within 1–1.5 m
Vanhems et al., 2013 [55]	F2F-Ps ^5^: number and duration of F2F-Ps ^5^	RFID technology: exchanging of ultra-low-power radio packets	Active RFID badge, worn with a lanyard on the chest ^1^	Professional staff and patients at an acute care geriatric unit of a university hospital	Comfortable, unobtrusive High-resolution data recordings High temporal resolution of 20 s	F2F-Ps ^5^ detected only if the devices are within 1.5 m
Yu et al., 2015 [56]	Proximity: ND-Ps ^6^	Bluetooth module	Sociometric badge, worn around the neck ^1^	Simulated hand-off scenarios at an emergency care environment	Comfortable, lightweight, real-time measurement, unobtrusive	The reliability of ND-Ps ^6^ depends on distance between badges; it decreases with increasing distance
Obadia et al., 2015 [65]	F2F-Ps ^5^: number and duration of F2F-Ps ^5^	RFID technology: exchanging of low-power radio packets	Wireless sensor that the healthcare workers keep in the overcoat pocket and the patients keep in a pocket, or wear as a watch or around the ankle ^1^	Patients and healthcare workers in a hospital in northern France	Comfortable, unobtrusive, real-time measurement High-resolution data recordings	F2F-Ps ^5^ detected only if the devices are within 1.5 m Temporal resolution of 30 s
Yu et al., 2016 [35]	Proximity: detection vs. no detection and D-P ^7^	Both IR and Bluetooth sensors	Sociometric Badge, worn around the neck ^1^	Simulated team communication and patient care scenarios at an emergency department’s pediatric ward	Comfortable, unobtrusive, real-time measurement	The reliability of D-P ^7^ decreases with increasing distance between badges
Stefanini et al., 2020 [39]	Proximity: D-Ps ^7^	Bluetooth module	Sociometric Badge, worn around the neck ^1^	Surgical team of the Breast Unit of an Italian university hospital	Comfortable, unobtrusive, real-time measurement	The reliability of D-Ps ^7^ decreases with increasing distance between badges
Stefanini et al., 2021 [40]	Proximity: D-Ps ^7^	Bluetooth module	Sociometric Badge, worn around the neck ^1^	Doctors and nurses of an emergency department of a hospital	Comfortable, lightweight, unobtrusive, real-time measurement	The reliability of D-Ps ^7^ depends on distance between badges; it decreases with increasing distance
Isella et al., 2011 [24]	F2F-Ps ^5^: number and duration of F2F-Ps ^5^	RFID technology: exchanging of ultra-low-power radio packets	Active RFID badge ^1^	Healthcare personnel, patients, and their caregivers at the pediatric ward of a hospital to study infectious disease transmissions	Comfortable, easy-to-wear, lightweight, unobtrusive High-resolution data recordings High temporal resolution of 20 s	F2F-Ps ^5^ detected only if the devices are within 1–1.5 m
Olguin D. et al., 2009 [36]	F2F-Ps ^5^: duration of F2F-Ps ^5^	RFID technology: exchanging of power radio packets	Sociometric Badge, worn around the neck ^1^	Nurses of a post-anaesthesia care unit (PACU)	Comfortable, lightweight, unobtrusive High-resolution data recordings	F2F-Ps ^5^ detected only if the devices are within 3 m
Endedijk M. et al., 2018 [38]	Speech activity: proportion of ST ^8^ proportion of OS ^9^ conversational imbalance in speech ^10^ proportion of ST ^8^ of the DS ^11^	Microphone	Sociometric Badge, worn around the neck ^1^	Master’s students of a ‘Technical Medicine’ Master’s program during simulated medical emergencies	Comfortable, lightweight, real-time measurement No CCR ^12^	Computational complexity issues
Yu et al., 2016 [35]	Speech activity: ST ^8^ OS ^9^ total ST ^8^: ST ^8^ and OS ^9^ for all the participants identification of the DS ^11^	Microphone	Sociometric Badge, worn around the neck ^1^	Simulated procedures of care assistance at an emergency department’s pediatric ward	Comfortable, lightweight, real-time measurement No CCR ^12^	Compared to video analysis, the sociometric badges appear to underestimate ST ^8^
Stefanini et al., 2020 [39]	Speech activity: %ST ^13^ %OS ^14^ %SLT ^15^ VN-PS ^16^ C-SA ^17^	Microphone	Sociometric Badge, worn around the neck ^1^	Surgical team at a university hospital	Comfortable, lightweight, real-time measurement No CCR ^12^	
Stefanini et al., 2021 [40]	Speech activity: %ST ^13^ %OS ^14^ P-AP ^18^ A-MF ^19^ A-MB ^20^	Microphone	Sociometric Badge, worn around the neck ^1^	Doctors and nurses of an emergency department of a hospital	Comfortable, lightweight, real-time measurement No CCR ^12^	
Olguin D. et al., 2009 [36]	Speech activity: A-VM ^21^ STD-VM ^22^ ST ^8^	Microphone	Sociometric Badge, worn around the neck ^1^	Nurses of a post-anaesthesia care unit (PACU)	Comfortable, lightweight, real-time measurement No CCR ^12^	

^1^ It can detect the parameter of only individuals wearing the systems; ^2^ T-F2F: total amount of F2F time; ^3^ NP-F2F: number of different people with F2F; ^4^ N-F2F: total amount of F2Fs; ^5^ F2F-Ps: F2F proximity interactions; ^6^ ND-Ps: number and duration of close proximity detections; ^7^ D-P: duration of each close proximity detection; ^8^ ST: speaking time; ^9^ OS: overlapping speech; ^10^ the standard deviation of ST7 of each team member, corrected for the duration of the relevant time window; ^11^ DS: dominant speaker; ^12^ CCR: conversation content recordings; ^13^ %ST: percentage of speaking time; ^14^ %OS: percentage of overlapping speech; ^15^ %SLT: percentage of silence time; ^16^ VN-PS: variation in the number of participants speaking, normalized by time and number of group members; ^17^ C-SA: consistency of each badge’s speech amplitude; ^18^ P-AP: the proportion of active participation within the conversation; ^19^ A-MF: audio acquired by the microphone at the front of the device; ^20^ A-MB: audio recorded by the back microphone; ^21^ A-VM: average volume modulation; ^22^ STD-VM: volume modulation’s standard deviation.

## 3. Wearables for Monitoring Individual Traits

The OR environment can be highly stressful, with surgeons, nurses, anesthesiologists, and other team members working under time constraints and demanding conditions [77]. Monitoring individual traits such as HR, HRV, RR, GSR, and the variation in physical activity level can be crucial in investigating CI in the OR due to their potential impact on individual and team performance [77]. Firstly, the above-mentioned parameters are closely linked to stress levels and emotional states. By their monitoring, it is possible to identify moments of heightened stress or emotional arousal among team members, which can help assess their impact on CI and performance. Secondly, they can be indicators of cognitive load and attentional focus. During complex surgical procedures, the cognitive demands on team members can be high, and the physiological traits provide insights into the mental workload experienced by individuals and how it varies throughout the procedure [78]. Moreover, prolonged surgical procedures can lead to fatigue among team members, which can impact performance and decision-making [79]. 

Measuring individual traits can help identify signs of fatigue by tracking changes in baseline HR, irregularities, or patterns indicative of exhaustion [77]. This information can guide scheduling practices, rest breaks, and workload management to maintain optimal performance and prevent errors. By integrating this physiological data with other observational and outcome measures, researchers can gain a more comprehensive understanding of the factors influencing surgical team effectiveness and identify strategies to optimize team performance and patient outcomes in the OR. 

Several studies used wearable technology to extract individual traits of the medical team during the work shift [80,81,82,83,84,85]. 

All the individual parameters that can be relevant for unveiling CI in clinical settings are shown below. The reasons why each feature is so useful for the analysis, as well as the technology utilized to evaluate each one, will be described. In particular, the sensors embedded into wearables and their body locations are shown in Figure 2. 

A schematic overview of the primary studies focusing on the use of wearables for monitoring individual traits is reported in Table 2. 

### 3.1. Heart Rate

HR is a fundamental physiological parameter that provides insights into the cardiovascular system’s functioning and overall health. It is expressed as the number of beats per minute (bpm) and it typically ranges between 60 bpm and 100 bpm in a healthy adult at rest [86]. 

Monitoring HR gives important details about one’s overall health, stress levels, and exercise intensity as well as cardiovascular health. It enables the evaluation of responses to physical activity, the tracking of changes over time, and the evaluation of recovery patterns.

Various methodologies, depending on the desired level of accuracy, context, and available equipment, exist for monitoring cardiac activity [86]. The majority of wearable devices to monitor cardiac activity are based on the following approaches: (1) phonocardiography (PCG), which identifies heart and blood flow sounds [87]; (2) photopletismography (PPG), which is an optical method for assessing blood volume changes [88]; (3) electrocardiography (ECG), which measures the electrical activity of the heart [89]; (4) the ballistocardiogram (BCG), which detects the recoil forces of the body in response to heart pumping blood into the vessels [90,91]; (5) the seismocardiogram (SCG), which represents the chest wall vibrations due to the heartbeat [90,92]; (6) impedance cardiography (ICG), which emerged for detecting the variations in thoracic impedance caused by the changing amount of fluid in the chest [93]; and (7) gyrocardiography (GCG), which measures the heart’s movements using gyroscopes [94].

The most popular technique is the ECG. It is a non-invasive, surface-based recording that shows an electrical tracing of the heart. To date, wearable technologies are currently having a rising impact on ECG signal monitoring. Utilizing cutting-edge communication protocols, smart and advanced wearable monitoring systems gather and transmit biomedical signals over long distances. For using such devices properly, applicability, reliability, and accuracy issues should be taken into account [95,96]. The integration of wearable technology, wireless sensor networks, and artificial intelligence applications can lead to novel approaches able to enhance healthcare delivery.

Recent advancements in wireless ECG monitoring have led to the spread of textile-based wearable devices.

Some studies focused on ECG tracking by using smart garments [97]. Wearable biomedical sensors can be directly placed on the patient or embedded into wearable clothing with electrodes. In particular, wireless sensors for monitoring physiological traits are quickly developing technologies for raising the standard of care while cutting costs. LOBIN [98] represents an ECG monitoring system which consists of a fusion of e-textiles and a wireless sensor network (WSN) embedded into a smart shirt. Another innovative ECG wearable device is proposed in [99] for extended duration measurements. Moreover, Blue Box [100] is a wireless handheld device developed for gathering and transmitting ECG, PPG, and bio-impedance parameters. 

As for PPG, measurements are frequently taken without any invasiveness at the skin’s surface [88]. PPG operates at a low-intensity IR light. The technique is used in commercially accessible medical devices as pulse oximeters, vascular diagnostics, and digital beat-to-beat blood pressure measurement systems and has a wide range of clinical applications [88]. The basic PPG device just needs a few opto-electronic elements, including a light source and a photodetector to track fluctuations in light intensity in response to blood volume variations [88].

The PPG waveform comprises a pulsatile waveform, frequently named the ‘AC’ component, and has a fundamental frequency around 1 Hz, depending on HR [88]. HR is computed by detecting peaks in the PPG signal. The principle behind this approach is that light, when passing through biological tissues, is absorbed differently [88]. In particular, light is absorbed more intensely by the blood than adjacent tissues, so PPG sensors can exploit the variations in light intensity to evaluate modifications in blood flow. The PPG voltage signal indicates the quantity of blood entering the vessels. Modern PPG sensors frequently employ semiconductor technology with LED and photodetector devices that use the red and/or near infrared wavelengths [88].

BCG and SGC can allow the evaluation of HR in everyday scenarios thanks to their easy application [101]. Today, technological improvements facilitate the application of BCG and SGC and lead to new frontiers in their clinical use. SCG-based non-invasive approaches are also gaining traction in assessing cardiac activity for at-home applications. In particular, some reviews on SCG are available for retrieving information about advances in this field [90,102]. Nowadays, the SCG signal can be acquired by using an accelerometer placed on the chest [86,103]. When choosing a tri-axial accelerometer, three specific SCG components are provided [104]. Nevertheless, most research only refers to the magnitude of the dorsoventral component [90]. Moreover, the SCG monitoring based on FBG sensors likewise results in being widely spread [25,27,105]. 

Additionally, the BCG signal can be another medium for the interpretation of heart activities [106]. The BCG can be quantified as a displacement, velocity, or acceleration signal [90]. HR is extracted by the intervals between consecutive J-wave peaks in the BCG signal, which corresponds to the movement of ventricular valves [86,90,101]. Various algorithms are available to extract the J-peaks and HR from BCG signals [90]. Since 2000, studies on BCG’s medical benefits have gained traction globally [90]. The possibility to continually collecting data during daily activities is one of the main benefits of wearables based on BCG or SCG [103,107].

The three-axis accelerometer is a sensor type widely used for wearable BCG or SCG measurements. It can be mechanically attached to the body using either adhesives, plastic mounts, or fabrics. 

Wearable technology allows for recording in any settings, giving researchers the chance to evaluate an individual’s cardiovascular performance in response to diverse environmental conditions or stressors. Considering highly demanding clinical scenarios, especially the OR, many studies examine the impact of the stress experienced by physicians throughout surgical procedures on the cardiovascular system by measuring HR and HRV. Chronic stress can have consequences on their health status, but also on surgical performance and patient safety [108].

Medical teams were frequently observed in simulated rather than real-world scenarios due to the complexity of the clinical context, as was revealed in [38,109]. However, although limited in number, applications in real scenarios were carried out. The most common type of wearable device used is a chest strap to be placed around the thorax, which includes dry electrodes for ECG tracking. In particular, a single-lead ECG trace is frequently recorded using the Zephyr BioHarness (Medtronic, Eindhoven, The Netherlands), which embeds two dry conductive electrodes [5,85]. In [108], surgical teams were monitored by using the Equivital sensor system EQ-01 (Hidalgo Ltd., Cambridge, UK), which consists of a chest belt with three dry electrodes to extract HR and HRV from a three-channel ECG-recording. Moreover, in [110], the feasibility assessment of a watch worn at the upper arm level was provided to monitor HR of surgeons during robotic-assisted procedures in OR and the findings appear promising in terms of accuracy. 

The aforementioned wearable technologies have the benefit of being comfortable and lightweight devices that do not hinder the medical team’s movements.

Additionally, a quantitative analysis using wearable technology is sometimes integrated with a qualitative analysis that leverages self-reports to track the level of stress experienced by the surgical team. One of the most widespread questionnaires is the State-Trait Anxiety Inventory (STAI), developed in the English language in 1970 by Spielberg et al. [111]. It is designed to assess an individual’s anxiety level by measuring both temporary state anxiety and long-standing trait anxiety. 

One factor that further complicates the monitoring of such a team is very often the impossibility of involving all the group members in the acquisitions [108], since they are not always willing to sign informed consent.

### 3.2. Heart Rate Variability

HRV is expressed as the changes over time of the intervals between successive heartbeats [112]. The heart’s capacity to adjust to varying conditions is considered to be reflected by HRV, which can detect and react quickly to unforeseen stimuli. HRV analysis allows for evaluating the condition of the autonomic nervous system (ANS), which controls heart activity and overall cardiac health. HRV is a valuable signal for gaining insights into the state of the ANS. The natural fluctuations in HR are a consequence of the autonomic neural control of the heart and the circulatory system [112]. The sympathetic nervous system (SNS) and parasympathetic nervous system (PNS) work in balance to control HR. Cardio-acceleration arises from increased SNS or decreased PNS activity. On the other hand, cardio-deceleration results from low SNS activity or high PNS activity. The level of HRV provides information about how effectively the nervous system regulates the HR and the responsiveness of the heart. HRV is valuable for determining the SNS and PNS functioning [112]. From a detailed analysis of the ECG, it is possible to discover if high HRV values are caused by diseases like atrial fibrillation [113]. Ideal HRV levels demonstrate an intact self-regulatory capacity [113].

If utilized in accordance with defined procedures, HRV measurements may be simple, non-invasive, and have good reproducibility. There exist various typologies of metrics, especially defined in time-domain, frequency domain, and non-linear metrics [113]. Firstly, the time-domain metrics assess the extent of variability in the interbeat intervals (IBI), which are defined as the time periods between consecutive heartbeats. According to [113], the typologies of time-domain indices are the SDNN, SDRR, SDANN, SDNN Index, RMSSD, NN50, pNN50, HR Max—HR Min, the HRV triangular index (HTI), and the Triangular Interpolation of the NN Interval Histogram. For instance, the root mean square of successive differences between normal heartbeats (RMSSD) is computed by first calculating each IBI in ms. Subsequently, before obtaining the square root of the total, every value is squared and then the result averaged. The RMSSD is indicative of the PNS responses under normal breathing conditions. A lower RMSSD value corresponds to a higher level of physiological arousal [112]. Secondly, the frequency-domain metrics can be used to characterize HRV. They provide an assessment of the distribution of absolute or relative power into four frequency bands: ultra-low frequency (ULF), very-low-frequency (VLF), low-frequency (LF), and high-frequency (HF) bands [113]. Finally, non-linear measurements are sometimes carried out. 

As for HR monitoring, HRV can be extracted from different typologies of signals (e.g., the ECG, PPG, BSG, and SCG signals), by evaluating time domain, frequency domain, and nonlinear domain HRV metrics, as shown above. 

HRV can provide useful information about the level of stress and workload of medical and OR teams [21]. In most cases, its monitoring is combined with that of HR [5,108,114]. Some experiments used T-shirts with embedded microelectronics, such as the VitalJacket^®^ (Biodevices Setúbal, Portugal S.A), that represents a wearable and certified medical device. In [21], this system has been used to monitor the stress and fatigue among two neurosurgeons during intracranial aneurism procedures. Moreover, as previously reported for HR measurement, the Zephyr BioHarness (Medtronic, The Netherlands) has been widely used [5,108,114], thanks to its light weight and high level of comfort for the users. 

### 3.3. Respiratory Rate 

Breathing is one of the physiological processes of the human body. It ensures the control of the acid–base equilibrium, gas flows, and other homeostatic mechanisms. Normal RR values, usually expressed in breaths/min, can vary from 12 breaths/min to 20 breaths/min in adults [115].

RR can be affected by a variety of pathological disorders (e.g., adverse cardiac events and clinical dysfunctions) and stressors (e.g., emotional stress, mental strain, physical burden) [116] A growing body of evidence suggests that RR is a primary trait to monitor in various scenarios [116]. In clinical settings, RR aids in the diagnosis of acute pneumonia, forecasts cardiac arrest, and offers information on clinical course [117,118]. RR is also a reliable predictor of physical fatigue during exercise [119] and it is related to exercise tolerance. Emotions have been shown to have an impact on ventilation [120]; consequently, RR appears to be a valuable feature in detecting emotional states in various scenarios. A rise in RR is shown, for example, during panic attacks [121]. The correlation between RR and the emotional sphere is due to the fact that RR is partly controlled by brain regions responsible for emotional processing [122]. Specifically, the amygdala, when directly stimulated, can lead to an increase in RR. 

In light of the aforementioned factors, it is clear how monitoring the RR of surgical team members can provide valuable insights into the CI and overall performance of the OR team. For assessing RR, the unobtrusiveness is an essential requirement, even more in the critical OR scenario, as intrusive systems may affect the emotional state and ventilatory processes. Clearly, RR is just one of the features that can aid in identifying emotions. Other indicators include HR, HRV, GSR, body temperature, body posture, and facial expressions [123,124].

Advancements in sensor technology for measuring RR have experienced rapid growth and some innovative solutions are proposed [125]. Existing devices are distinguished according to their working principle and mode of use, and they are classified as contact or non-contact systems. In the first case, the devices are placed directly in contact with the participant’s skin, as opposed to the second case. 

Non-contact systems have the benefit of enhancing comfort, which appears essential for long-term monitoring, and providing more accurate measurements, since the body contact may hinder the user’s movements, thus producing alterations in RR values [116].

In particular, most contact devices measure parameters like respiratory sounds, respiratory airflow, respiratory-related chest or abdominal movements, respiratory CO_2_ emission, and oximetry probe SpO2 [126]. Moreover, the RR can also be derived from the ECG. Acoustic-based methods measure respiratory sounds by using a microphone that is either positioned directly over the throat or close to the respiratory airways to listen for sound variations. They exploit frequency domain analysis and evaluate the sound intensity. Several miniaturized and wearable respiration monitoring systems have been proposed [127], also for detecting sleep apnea in infants [128]. Moreover, other methods are based on different properties of exhaled and inhaled air, since exhaled air is warmer, wetter, and richer in CO_2_ than inhaled air. The RR can be determined via these variations. Most airflow-based techniques require a sensor to be connected to the airways [129], such as a nasal or oronasal thermistor [130], a nasal pressure transducer [131], nasal cannulae, a mouthpiece, or a facemask. Additionally, chest and abdominal movement detection for estimating RR is also widely spread. In this approach, strain gauges or impedance methods are mainly used [116]. The transcutaneous CO_2_ monitoring is another available technique, consisting of an electrode that is heated to approximately 42 °C and placed on the skin [116]. Blood oxygen saturation (SpO_2_) estimation is another method for keeping track of the effects of irregular ventilation. It is based on the estimation of blood saturation, also called SpO2, which represents the proportion of oxygen-saturated hemoglobin in the blood [132]. A pulse oximeter that uses red and infrared frequencies is typically employed. Finally, from the literature, RR measurement derived from ECG also emerged. This approach relies on the observation that respiration can affect the ECG trace. Therefore, the ECG fluctuation can be measured for determining the RR [133].

In addition to this, there exist contactless techniques suitable for monitoring RR. One of them is represented by a thermal camera that can be utilized for extracting RR from thermal video frames. This technique allows for performing RR estimation by examining changes in pixel intensity caused by respiration process [134]. In contrast to most contact-based approaches, video picture post-processing is typically time-consuming, and IR video images are typically analyzed after data collection. RGB camera sensors and depth sensors are additional contactless sensors that can simultaneously record RR, facial emotions, and cardiovascular data [135,136,137]. 

Research on RR monitoring in clinical settings is emerging from the literature. In [5], RR monitoring is carried out by using the Zephyr BioHarness (Medtronic, The Netherlands), which consists of a chest strap to be positioned around the thorax. It integrates a conductive textile to record the breathing waveform by the chest wall excursions. An anesthesiologist and a medical trainee were involved in this study during an epidural procedure on a patient afflicted by chronic back pain. The BioHarness was selected because of its many benefits for the clinical settings in terms of performance and reliability. Firstly, the device can collect data for 26 h, meaning it is durable. It is wearable by both men and women, with a mass of 85 g [138]. Moreover, there is compelling evidence that demonstrates its reliability across multiple contexts [139]. 

### 3.4. Galvanic Skin Response 

The GSR represents a modification in the electrical properties of the skin. As a measure of sweat gland function, the signal can be utilized for detecting autonomic nerve responses [140]. The response is typically observed as an increment in the skin’s electrical conductance and, consequently, a reduction in resistance. It appears about 2 s after stimulation, reaches its peak within 10 s, and then decreases at about the same rate [140]. 

The GSR has been extensively used in research concerning emotions and emotional learning because it has a higher sensitivity than other physiological reactions [141]. An experienced professional can determine which stimuli cause emotional disturbance by observing when the response takes place. The GSR is essentially involuntary, but through biofeedback training, people can learn to somewhat control it [142]. 

The time-dependent GSR signal can be analyzed in both the time and frequency domains. In the first case, two GSR components are typically extracted: the Skin Conductance Level (SCL) and Skin Conductance Responses (SCR) [143]. The SCL represents the tonic component, which varies slowly over time. Several external and internal variables can influence SCL (e.g., psychological parameters and skin properties). It reflects the level of nervous system activation in an individual even when there are no external stimuli or environmental events. SCL can exhibit significant variations among individuals, according to their hydration skin levels and autonomic regulation. SCL analysis alone is not sufficient [144], so an SCR examination is often added. The SCR represents the rapid fluctuations due to the SNS reaction to a stimulus. This phasic component can be classified as event-related SCR or nonspecific SCR. The first one refers to the response to certain events or external factors, such as sounds and images, whereas the other one represents signal modifications induced by internal stimuli, like thoughts, memories, and emotions. From the SCR signal, it is possible to extract the number of peaks that typically exceed an amplitude threshold set at 0.01µS. In particular, the number of SCR peaks is subject-specific [144]. Considering the differences between SCL and SCR, each component is often examined independently after using a time-domain subtraction algorithm to isolate the SCR from the SCL. 

Several parameters can be extracted from the SCR: the latency, which refers to the time interval between the stimulus and peak starting point; the peak amplitude; the rise time, which is the duration between the onset and the peak; and the recovery time, which refers to the period from peak to complete recovery. 

GSR examining can be challenging, mainly because it is highly variable depending on the subjects investigated, and it can be affected by factors like motion artefacts [145]. For this reason, frequency domain analysis is also sometimes implemented by computing the Power Spectral Density (PSD) [146]. It offers fewer but more reliable details, especially if the user is exposed to specific types of stimuli [145]. 

GSR is generally assessed by using non-invasive methods, which involve applying two electrodes to the skin. GSR sensors include electrodes that can be of various types. The Ag/AgCl electrodes are the most commonly used, but alternative dry carbon/salt adhesive electrodes or textile Ag/AgCl type electrodes can be employed. In some cases, a gel containing NaCl or KCl is added between the skin and electrodes to minimize the impedance between them [145].

Certain electrode application sites are found to produce more intensive responses to arousal, due to the higher concentration of sweat glands. Among them, the fingers, the palmar regions, the foot, the wrist, and the shoulder emerge. 

There are primarily three distinct methods for assessing the GSR: (i) the endosomatic method, which does not involve the application of an external current, (ii) the exosomatic technique using Direct Current (DC), and (iii) the exosomatic technique using Alternative Current (AC) [147]. As for the endosomatic measurement, two electrodes are used, one of which is the reference electrode located on a not highly active region, like the forearm. 

Over the years, several systems have been proposed to measure the GSR signal [148]. 

These are often developed to assess other physiological traits as well, such as HR, body temperature, and blood pressure, with the intention of providing more accurate insights into the participant’s emotional engagement [149].

A review of the literature reveals the significance of GSR signal monitoring in the clinical setting. Monitoring the degree of stress and cognitive burden experienced by the surgical team members is particularly important because it can help prevent medical blunders. In the majority of applications, simulated scenarios—in particular, those depicting medical emergencies or surgical procedures—are chosen. Some studies apply GSR sensors on the wrists and ankles of surgeons during simulated medical procedures, since this is less demanding for surgical team members [80,81,82,83]. In [38], a relatively unobtrusive wristband, named Empatica E4, was employed. It included 8 mm silver-plated electrodes placed on the wrist of the team members’ non-dominant hands. Moreover, in [82], a GSR device (Manufactured by Neumitra, Inc., Boston, MA, USA), containing two silver chloride electrodes placed on the wrist and ankle was used to acquire the GSR signal. 

Currently, only a few studies have focused on measurements in real scenarios [150,151], probably due to the placement of the electrodes, which can risk hindering the movements of the surgeon and their team. In [150], a wearable device was used to monitor expert and novice surgeons during 21 surgical procedures for studying the effect of surgical outages on the surgeons’ cognitive, emotional, and physiological states. Furthermore, in [151], a GSR sensor (Affectiva Q Sensor, Affectiva Inc., Waltham, MA, USA) was employed with two dry 12 mm electrodes spaced 4 mm apart, positioned on both wrists of the participants without adding any conductive gels or pastes. To guarantee sterility during laparoscopic cholecystectomies in OR, it was not possible to position the sensors on the palms or fingers, as is frequently performed with GSR studies. 

### 3.5. Physical Activity Level

The level of physical activity is another helpful indicator for examining collective human behavior [34]. Some research shows that the human gait is a distinctive trait, which can also be useful for robust identity recognition [152].

Variation in physical activity level can influence the group outcomes [18], adding useful information in the evaluation of CI also in the clinical field. Indeed, from the literature, it emerged that activity level is associated with an individual’s level of concentration: the lower the level of physical activity, the higher the individual’s concentration [33]. 

Nowadays, several typologies of wearable devices are developed for monitoring physical activity level (e.g., sociometric wearable devices [17]), and, in most cases, they are based on embedded three-axis accelerometers [33,35,36,37,39,40,56,153].

By using the Sociometric Badge, considering that the range of human motion behavior is contained below 15 Hz, the three-axis accelerometer signal is sampled at fs ≥ 30 Hz [34]. Several different features can be extracted to characterize physical activity level. Knowing the acceleration signal vector magnitude, every minute the Sociometric Badge computes the average signal amplitude, the standard deviation of signal amplitude, and, finally, the power or signal energy [36]. The validation study [70] shows that the accelerometer embedded in each badge reliably captures the front–back and left–right tilting that are used to deduce posture. 

In the Business Microscope [33], the acceleration signal is recorded for 2 s every 10 s at 50 Hz, together with the motion rhythm, obtained by the number of wave zero-crossings of the acceleration signal. Additionally, it is possible to identify the behavior from the motion rhythm (e.g., 0 Hz is associated with sleeping, the range 1–2 Hz with talking, and so on). 

The Sociometric Badges and the Business Microscopes simultaneously record other individual and group parameters as described above. 

**Table 2 sensors-23-09777-t002:** A schematic overview of the primary studies focusing on the use of wearables for monitoring individual traits.

Paper	Parameter	Working Principle	Wearable	Scenario	Pros	Cons
Rieger et al. [108]	HR, HRV	3-channel ECG-recording	The Equivital sensor system EQ-01 (Hidalgo Ltd., Cambridge, UK) ^1^	Intraoperative monitoring of 20 surgeons, 6 residents, 5 fellows, 5 attending, and 4 chiefs of medicine to assess surgeons’ stress level	Comfortable and lightweight device AW ^2^	Off-line-SA ^3^ Individual physiological baseline acquired during resting period at night
Lo Presti et al. [5]	HR, HRV	Single-lead ECG trace	Zephyr BioHarness (Medtronic, The Netherlands) ^4^	Monitoring of an anesthesiologist and a medical trainee during the execution of an epidural procedure on a patient afflicted by chronic back pain	Comfortable and lightweight device AW ^2^	Off-line-SA ^3^
Joseph et al., 2016 [85]	HR, HRV	Single-lead ECG trace	Zephyr BioHarness (Medtronic, The Netherlands) ^4^	Monitoring of a trauma team composed of an attending trauma surgeon, a junior trainee, and a senior trainee during trauma activation and emergency surgeries	Comfortable and lightweight device AW ^2^	Off-line-SA ^3^ A few surgeons refused to be monitored
Lo Presti et al. [109]	HR, HRV	Single-lead ECG trace	Zephyr BioHarness (Medtronic, The Netherlands) ^4^	Monitoring of a subject invited to engage in unrestricted upper body motions to replicate common actions performed in OR	Comfortable and lightweight device	SW ^5^ Off-line -SA ^3^
Pimentel et al., 2019 [21]	HRV	Single-lead ECG trace	VitalJacket^®^ (Biodevices, Setubal, Portugal S.A) ^6^	Monitoring of stress and fatigue among 2 neurosurgeons during intracranial aneurism procedures	Comfortable and lightweight device AW ^2^	Off-line-SA ^3^ Limited involved population and limited number of acquisitions
Yamada et al. [110]	HR	Photopletismography	Apple Watch Series 8 worn on upper arm	Monitoring of surgeons during robotic-assisted surgery	Comfortable and lightweight device AW ^2^	Limited sample size
Lo Presti et al. [5]	RR	Breathing waveform by the chest wall excursions	Zephyr BioHarness (Medtronic, The Netherlands) ^4^	Monitoring of an anesthesiologist and a medical trainee during the execution of an epidural procedure on a patient afflicted by chronic back pain	Comfortable and lightweight device AW ^2^	Off-line-SA ^3^
Lo Presti et al. [109]	RR	Breathing waveform by the chest wall excursions	Zephyr BioHarness (Medtronic, The Netherlands) ^4^	Monitoring of a subject invited to engage in unrestricted upper body motions to replicate common actions performed in OR	Comfortable and lightweight device	SW ^5^ Off-line-SA ^3^
Endedijk et al., 2018 [38]	GSR: SCR signal, N-SCR-Ps ^7^ and A-SCR-Ps ^8^	GSR signal	Empatica E4 ^9^	Monitoring of Master’s students of the ‘Technical Medicine’ Master’s program during simulated medical emergencies	Comfortable, unobtrusive, and lightweight device	SW ^5^ Off-line-SA ^3^ Small sample size Signal baselines not acquired
Phitayakorn et al., 2015 [81]	GSR	GSR signal	GSR device (Manufactured by Neumitra, Inc, Boston, MA, USA) ^10^	Monitoring of 17 OR teams, composed by 2 anesthesiology residents, 2 general surgery residents and 2 practicing OR nurses during high-fidelity surgical simulations	Comfortable and lightweight device	SW ^5^ Off-line-SA ^3^ Possible falsely elevated GSR readings due to the use of surgical gowns and gloves
Lo Presti et al. [109]	GSR	GSR signal	Shimmer GSR+ sensor (Shimmer sensing, Dublin, Ireland) applying two electrodes on two fingers of the subject	Monitoring of a subject invited to engage in unrestricted upper body motions to replicate common actions performed in OR	Comfortable and lightweight device	SW ^5^ Off-line-SA ^3^
Van Houwelingen et al., 2020 [150]	GSR	GSR signal	SenseWear Pro 3 armband	Monitoring of expert and novice surgeons during 21 surgical procedures to study the effect of surgical flow irregularities on their cognitive, emotional, and physiological state	Comfortable device AW ^2^	Off-line-SA ^3^ The total number of involved surgeons is limited to 8, and most of them are experts
Jacob et al., 2017 [151]	GSR, SCL, SCR	GSR signal	A GSR sensor (Affectiva Q Sensor, Affectiva Inc., Waltham, MA, USA) ^11^	Monitoring of 14 general surgery residents during laparoscopic cholecystectomy	Comfortable and lightweight device Tiny size AW ^2^	Off-line-SA ^3^ Impossibility to position the electrodes on the palms or fingers to guarantee sterility in OR
Yu et al., 2015 [56]	Physical activity level: MAV ^12^ proportion of TM ^13^ L/R angle ^14^ F/B angle ^15^	Three-axis accelerometer	Sociometric badge, worn around the neck	Simulated hand-off scenarios at an emergency care environment	Comfortable, lightweight, real-time measurement 100% detection of the proportion of TM ^13^ in a static condition	Limited detection of TM ^13^ for walking participants SW ^5^
Yu et al., 2016 [35]	Physical activity level: MAV ^12^ proportion of TM^13^ L/R angle ^14^ F/B angle ^15^	Three-axis accelerometer	Sociometric badge, worn around the neck	Simulated patient care scenarios at an emergency department’s pediatric ward	Comfortable, lightweight, real-time measurement	• SW ^5^
Stefanini et al., 2020 [39]	Physical activity level: body movement ^16^ activity-walking ^17^ BMC ^18^ posture activity ^19^ PAC ^20^	Three-axis accelerometer	Sociometric Badge, worn around the neck	Surgical team of the Breast Unit of an Italian university hospital	Comfortable, lightweight, real-time measurement AW ^2^	-
Stefanini et al., 2021 [40]	Physical activity level: body movement ^16^ BMC ^18^ posture activity ^19^ PAC ^20^	Three-axis accelerometer	Sociometric Badge, worn around the neck	Doctors and nurses of an emergency department of a hospital	Comfortable, lightweight, real-time measurement AW ^2^	-
Olguin D. et al., 2009 [36]	Physical activity level: Signal amplitude’s average Signal amplitude’s standard deviation Power or signal energy	Three-axis accelerometer	Sociometric Badge, worn around the neck	Nurses of a post-anaesthesia care unit (PACU)	Comfortable, lightweight, real-time measurement AW ^2^	-
Rosen M. et al., 2018 [37]	Physical activity level: body movement ^16^ the overall activity ^21^	RFID technology: exchanging of power radio packets	Sociometric wearable badge	Nurses of a surgical intensive care unit (ICU)	Comfortable, lightweight, real-time measurement AW ^2^	-

^1^ It consists of a three-dry electrode chest belt; ^2^ AW: measurements performed during authentic work, not simulated work; ^3^ Off-line-SA: off-line signal analysis; ^4^ it consists of a chest strap which integrates two dry conductive electrodes; ^5^ SW: measurements performed during simulated work, not authentic work; ^6^ a wearable and certified medical device, consisting of a T-shirt with embedded microelectronics; ^7^ N-SCR-Ps: number of over-threshold peaks; ^8^ A-SCR-Ps: amplitude of over-threshold peaks; ^9^ a wristband with 8 mm electrodes in silver, placed on the wrist of the team members’ non-dominant hands; ^10^ it contains 2 silver chloride electrodes, placed on the wrist and ankle; ^11^ it includes 2 dry 12 mm electrodes spaced 4 mm apart, positioned on each user’s wrist; ^12^ MAV: mean angular velocity; ^13^ TM: time moving; ^14^ L/R angle: device’s angle in the left/right direction; ^15^ F/B angle: device’s angle in the forward/back direction; ^16^ energy amplitude over the three spatial directions; ^17^ percentage of time that the badge wearer was moving/walking; ^18^ BMC: body movement consistency; ^19^ absolute angular velocity; ^20^ PAC: posture activity consistency; ^21^ the absolute value of the first derivative of the magnitude of energy across the three-axis accelerometer sampled in 20 s intervals and averaged across an entire shift segment.

## 4. Discussions and Conclusions

In the present review, we provided a detailed overview of the key parameters in studying the group dynamics and social interactions for unveiling CI in a clinical scenario, since it appears essential to accurately analyze the behavioral dynamics and psychophysical state of surgeons, physicians, and nurses. We focused on wearable technologies to monitor behavioral and individual features due to their potentialities in overcoming the main issues related to approaches based on surveys and self-reports. Indeed, studies exploiting surveys to study human behavior and investigate people’s attitudes for leadership, communication, teamwork, stress, and fatigue have unluckily resulted in being inaccurate, time-consuming, and non-exhaustive [16].

In contrast, the monitoring of some behavioral and individual features that are reflective of CI has turned out to be beneficial in supporting CI assessment. However, only a few studies have focused on the use of wearables in this scenario, due to the criticality of the application context, and no one has yet employed different wearable technologies to simultaneously monitor the complete set of mentioned parameters.

In this work, we identified two categories of metrics that should be monitored simultaneously. Firstly, we introduced the group parameters, which are the F2Fs, proximity, and speaking time; secondly, we introduced the individual ones, which are HR, HRV, RR, GSR, and physical activity level. The relevance of each feature was highlighted, along with the wearable technologies used to evaluate each one in a clinical setting. Furthermore, we have summarized the different sensors’ working principles and the main advantages and disadvantages of each system (see Table 1 and Table 2), providing a comparison that can help researchers to choose the proper device for their specific applications. 

At the moment, due to the complexity of the issue, a fully developed and structured CI model has not yet been established. Indeed, although keeping track of the aforementioned parameters seems to be a promising method to unveil CI, the literature has not yet outlined valuable techniques for analyzing all the data collected. So far, only a few studies focused on this issue lay the foundations for future progress [154,155,156,157].

The next challenge will be to develop a shared CI model to promote new team training strategies, teamwork improvement techniques, and stress management procedures with consequent patient safety and surgical outcome improvements.

The proposed review is expected to be a first attempt to bring order to the unstructured CI field, and a guide for future CI investigations in medical scenarios. 

## Figures and Tables

**Figure 1 sensors-23-09777-f001:**
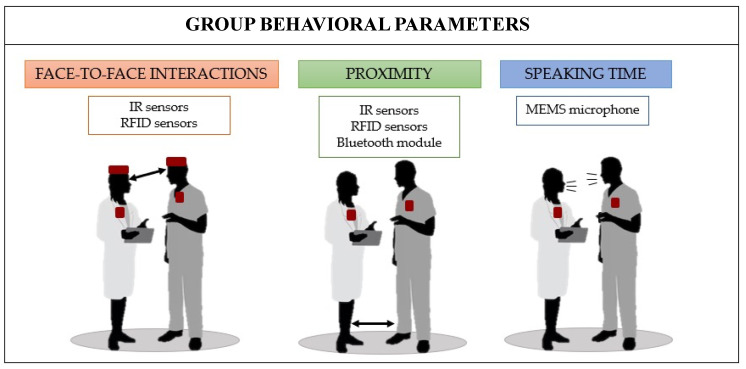
Sensors embedded into wearable devices used to monitor group behavioral parameters and their body location are shown. All these sensors have already been used in real/simulated clinical settings. The red parts highlight the possible position of the wearable sensors.

**Figure 2 sensors-23-09777-f002:**
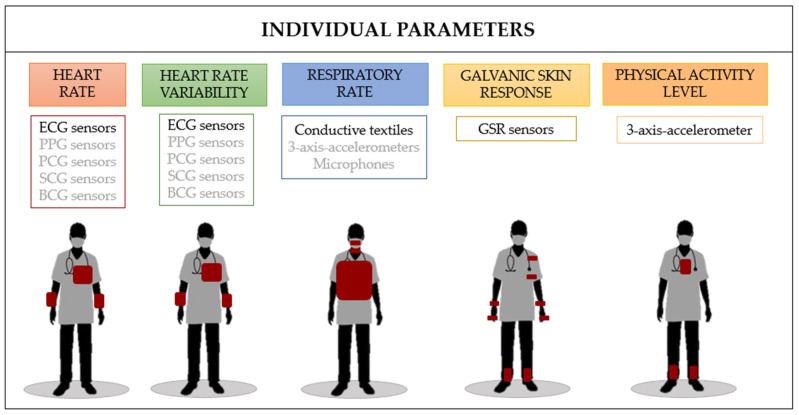
Sensors embedded into wearable devices used to monitor individual parameters and their body locations are shown. Those already used in real/simulated clinical settings are highlighted. The red parts highlight the possible position of the wearable sensors.

## Data Availability

Not applicable.

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
