# Peer review of "Wearable Systems for Unveiling Collective Intelligence in Clinical Settings"

_sensors, 2023, doi:10.3390/s23249777_

Round 1
Reviewer 1 Report
Comments and Suggestions for Authors
1. The structure which is slightly inappropriate, makes the order of the manuscript need to be adjusted, and some content is multifarious and delayed vaguely, making it laborious to form preliminary cognizance.
2. The essential characteristic parameters of how to evaluate CI which are mentioned in the first part of the manuscript are not highly correlated with the application in the medical field mentioned in the following part. Although the application scenarios are described in detail, there are still lacks of the connections with the essential parameters described in the previous sections. The technical summary is relatively comprehensive. However, there is no enough description of how to use these parameters for evaluating CI in plural occasions , which could be considered as another significant domain that can assist in fathoming how CI works in the realty.
3. The table structure need further rationalize.
4. The theme at the beginning of this manuscript needs to be further emphasized.
Reviewer 2 Report
Comments and Suggestions for Authors
One of the biggest issues with the readability of the review is the multipage tables 1 and 2 covering the various sensors. I would suggest that there should be a way to reduce the amount of text in these. For example where there are points that are repeated many times such as "It can detect F2F of only individuals wearing the systems." you have a key and use symbols or some other shorthand. I noticed that in table 1 for one of the papers, that one of the "Cons" was "Badges do not interfere with medical devices.", I would suggest that's a "Pro" and also that if there are any that DO interfere with medical devices they should be immediately rejected. I wonder also if the tables could be presented in landscape in a PDF version. Overall this is a great review of the various sensors, and while I might find the sociometric badge idea somewhat Orwellian I can see the point for limited studies. In addition this is a decent review of the different wearable sensors but I wonder if, in light of the fact that so many people wear watches that measure many of the parameters, that the current commercial wearable landscape should have been included. Similar criticism for Table 2, there has to be a better way to summarise the information.
Reviewer 3 Report
Comments and Suggestions for Authors
The manuscript presents a review on wearable devices and techniques for acquisition and analysis of signals from the human body that can be used for assessment of individual and group behavior and status. The manuscript is interesting and contains a lot of useful information. However, the authors have addressed a lot of studies that are not related to wearable systems applied in a clinical setting, so they should think about a title which is not so specific. Moreover, before publication, the authors should address the following questions, remarks and recommendations:
1) The authors have addressed a lot of studies, however, they have not disclosed the methodology for selection of these studies. Normally, when a literature review is planned, the authors set a time interval, language of the publications, scanned databases that contain scientific papers, keywords for the search, exclusion criteria, etc. Such approach guaranties that the authors would address all studies that met a pre-set criteria and the review would be unbiased. What are the inclusion/exclusion criteria applied for this review?
2) In Table 1 there are some studies that do not correspond to the title – i.e. they are not related to a clinical setting. These are 19, 22, 32, 33, 54, 55. Such studies should either be excluded from the table, or the authors should explain their relation to a clinical setting.
3) Related to the application of wearable ECG devices, issues such as applicability, reliability and accuracy of the applied portable personal ECG analyzers are worth to be mentioned (https://doi.org/10.1007/978-3-030-96638-6_33; https://ieeexplore.ieee.org/document/8878494).
4) In Table 2 reference 33 does not correspond to the title – i.e. it is not related to a clinical setting.
Are there studies that combine wearable measurement of group behavior and individual stress measurements? And what about studies on the influence of the individual stress on the group behavior – i.e. how the individual stress influence the collective intelligence? In section “Discussions and Conclusions” the authors have written: “Wearable technology can be a valuable solution for monitoring simultaneously both individual and group parameters without impairing the user activity.” Information about such studies should be provided within the text of the manuscript. What parameters are observed together?
Comments on the Quality of English LanguageMinor editing of English language required.
Round 2
Reviewer 3 Report
Comments and Suggestions for Authors
The authors have considered the recommendations in my first report and in my opinion the revised version of the manuscript could be published in its present form.